# Biomechanical Behavior Evaluation of Resin Cement with Different Elastic Modulus on Porcelain Laminate Veneer Restorations Using Micro-CT-Based Finite Element Analysis

**DOI:** 10.3390/ma16062378

**Published:** 2023-03-16

**Authors:** Meltem Mert Eren, Alper Tunga Celebi, Esra İçer, Cengiz Baykasoğlu, Ata Mugan, Taner Yücel, Esra Yıldız

**Affiliations:** 1Department of Restorative Dentistry, Faculty of Dentistry, Altınbas University, 34147 Istanbul, Türkiye; 2Institute of Applied Physics, Vienna University of Technology, 1040 Vienna, Austria; 3Department of Informatics, Technische Universität München, 85748 Garching, Germany; 4Faculty of Engineering Mechanical Engineering Department, Hitit University, 19030 Çorum, Türkiye; 5Mechanical Engineering Department, Faculty of Mechanical Engineering, Istanbul Technical University, 34437 Istanbul, Türkiye; 6Department of Restorative Dentistry, Faculty of Dentistry, Istanbul University, 34116 Istanbul, Türkiye

**Keywords:** porcelain laminate veneer, resin cement, laminate preparation, finite element method, micro-computed tomography

## Abstract

The aim of this study is to evaluate the biomechanical behavior of the porcelain laminate veneer restorations (PLV) of the maxillary central incisor luted with two types of resin cements having different incisal preparations: butt joint and palatal chamfer. Biomechanical analyses were performed using the micro-CT-based finite element models, and von Mises stress and strain values of the PLV, resin cement, adhesive layer, and tooth structure were computed. The PLV with butt joint preparation showed larger stress values than those of restored with palatal chamfer preparation, regardless of the elasticity of the cement and loading conditions. An increase in the elasticity modulus of the resin cement induced slightly larger stresses on the adhesive layer, tooth tissues, and restorative materials. Overall, this study demonstrates the role of the preparation design and luting materials on the mechanical behavior of the PLV restorations and discusses the potential failure mechanisms of the PLV restorations under different loading mechanisms.

## 1. Introduction

The use of porcelain laminate veneers (PLV) in restoring discolored, eroded, fractured, and disintegrated teeth has become increasingly popular in the past years due to their high performance against chewing forces [1]. Survival rates of PLV range from 82% to 96% in 10 to 21 years as expressed in clinical studies [2,3,4]. The longevity of PLV restorations depends on various factors, such as tooth structure, type of preparation, preparation depth, adhesive and resin cement, occlusal relation, and parafunctional behavior [5,6]. The preparation design is, therefore, essential in controlling over-contour, achieving a resilient adhesion, and preventing irregular stress distribution on restored teeth. In addition to the preparation design, the elasticity modulus difference of adhesive resin cements shows an impact on the long-term success of porcelain laminate veneers [6,7]. Due to the coverage ability of resin cements in the restoration assembly, minor defects and microcracks can be inhibited from propagating, and therefore, an increase of the fracture resistance is achieved.

The influence of the preparation design and the material properties on the clinical performance of PLV restorations has been studied earlier [8]. The adhered interfaces for ceramic veneers and conventional porcelain showed a debonding rate of 5.6–14% within 2.5–10 years of clinical performance [9]. Strong adhesive cementation ensures a sufficient bond strength between the tooth and the restorative material [9]. Although there are several studies reporting the strengthening effect of resin cement on the PLV restorations, information on the durability of the tooth-cement-restoration assembly is lacking [10,11,12,13,14]. As a result, it is unclear how the mechanical properties of cement material for the bonded ceramic veneer restoration change with respect to the strength of the resin cement interface.

Observation of critical loads on teeth or restorations and quantifying the stresses and deformations due to these loads are paramount to understand the ideal restorations for the clinical treatments. The finite element (FE) analysis is a commonly used computational technique in restorative dentistry for determining stresses, strains, and displacements that occur because of the functional forces on teeth and restorations [5,15]. Combined with realistic imaging and modelling techniques, FE analysis provides an accurate approach to examine stress/strain distributions in dental structures due to various external factors, such as mechanical forces, acidity, or thermal loads [16,17]. FE-based predictions aid in designing and developing materials against common failure mechanisms in restorative dentistry, such as fracture and delamination.

Simple geometries prepared via computer-aided design (CAD) are often used in FE analysis of dental structures and restorations. This approach indeed provides simplicity, but it does not include any anatomical irregularities or possible imperfections of the tooth, and also it creates difficulties in modelling multi-layered structures, such as PLV restorations. Modern scanning techniques, such as micro-computed tomography (micro-CT), cone-beam computed tomography, and optical scanning, enable high resolution three-dimensional (3D) images of dental tissues and restorations [18]. Micro-CT achieves an accurate surface and interior model of the dental components without causing any damage (i.e., non-invasive technology). Starting with a reliable natural tooth X-ray image, one can generate a precise geometric model for FE analysis. Since micro-CT obtains dental tissue (enamel and dentin) volume in detail for generating a volumetric mesh of the tooth structure, micro-CT-based realistic modeling presents a more accurate prediction of the biomechanical behavior of dental components. There are various dental structures of tooth layers, such as enamel, dentin, pulp chamber, and cement, which were achieved using micro-CT in earlier studies [19,20,21,22].

In the light of the information above, the main goal of this study is to evaluate the biomechanical behavior of PLV restoration in the maxillary central incisor with two incisal preparation types and with different adhesive cements under horizontal and oblique loads that mimic chewing forces. To our knowledge, there is no study in the literature examining the effect of cementation materials on the stress and strain distributions of the PLV restorations using FE analysis. Additionally, this study distinguishes by introducing precise computational models based on micro-CT images of the restored and sound tooth. Generated geometric models include each layer of the maxillary central incisor, PLV restoration, and the surrounding bones.

## 2. Materials and Methods

### 2.1. 3D Solid Model Generation

3D solid models were artificially created using a freshly extracted sound maxillary first central tooth. After taking an impression of the sound tooth (Heraform Type A + B, Heraeus Kulzer, Hanau, Germany), artificial models were casted using a polyurethane die material (AlphaDie MF, Schütz Dental GmbH, Rosbach, Germany). Artificial teeth were then applied to two different incisal preparation designs: butt-joint and palatal chamfer. First, butt joint preparation was achieved using a deep cutting bur (DC.5, DC0.3, 782.10C, 782.10VF- Two Striper, Abrasive Technology, Lewis Center, OH, USA). The same process was repeated, and then the butt joint prepared tooth was converted into the palatal chamfer preparation with additional steps [23].

Prepared artificial teeth were then scanned using Skyscan 1172 high-resolution micro-CT (Skyscan, Aartselaar, Belgium) to achieve 3D digitized models. The sound maxillary central tooth was scanned with a voxel dimension of 9.16 µm and with a total number of 2167 slices. Artificial models with butt-joint and palatal chamfer preparations were scanned with a voxel dimension of 18.86 µm and 15.14 µm, respectively. A total of 592 slices were obtained for two preparation designs. Then, all cross-sectional micro-CT image files were processed using an interactive medical image control system (MIMICS 14.12, Materialise, Leuven, Belgium). The dental tissues in the sound tooth were segmented based on their value of image density thresholding of the micro-CT data. Dentin and enamel point clouds were processed after image adjustment to obtain the macro models. Subsequently, the scanned geometries were processed with Computer Aided Design (CAD) software (SolidWorks Corp., Concord, MA, USA).

To generate solid models of each PLV design, restored teeth geometries were subtracted from the sound tooth geometry. Subsequently, PLV, resin cement, and adhesive layer were segmented using the subtracted geometry (Figure 1). The thickness of the adhesive layer and the resin cement were selected as 50 µm and 100 µm, respectively. A uniform thickness for both layers was assumed due to its simplicity in FE modelling. It should be noted that the resin cement thickness is not uniform in clinical practice, and it can be thinner (e.g., 15 µm) at the edges of the restorations compared to the body region. Based on the root-form geometry of the central incisor, periodontal ligament (PDL), and the spongious and cortical bones were created. The PDL was generated by thickening it outwards by as much as 200 µm. In addition, the dentin had been partially exposed in the cervical area during the tooth preparation, so these areas were carefully separated from the enamel, as shown in Figure 2. 

### 2.2. 3D FE Modeling

Linear tetrahedral solid elements were used in FE analysis. Convergence test was conducted by comparing the stress distributions around cervical and buccal areas. After the convergence test, the total number of elements and nodes were, respectively, 471,917 and 111,655 for the sound incisor model, 507,590 and 125,333 for the butt joint preparation, and 504,351 and 124,462 for the palatal chamfer preparation. Two different mechanical loads were applied to the palatal surface of each model. A 15 N force with a 90° angle between the tooth’s longitudinal axis and the normal vector of the palatal surface simulates the protrusive movements. Secondly, an oblique force with a 50 N magnitude and 60° angle represents the tearing function. These forces were placed at an approximate 2 mm distance to the incisal margin where they were distributed over multiple nodes (20 nodes) at the center of the palatal surface to avoid unrealistic stress concentration on a single node. The applied forces used in this study are illustrated in Figure 3a. Surrounding bonds play a vital role in terms of boundary conditions, significantly affecting the top coronal part. Therefore, the motion of the nodes on the outer surfaces of cortical and cancellous bones was restricted in all degrees of freedom to mimic real-life conditions (see Figure 3b). Modeling the PDL is also critical because it transfers force through the bones and avoids building up residual stresses occurring on dentin and enamel. It is critical to note that the contact region between all tissues and restorative materials was assumed to be perfectly attached and modeled using tie constraints. Such constraints fixed all translational and rotational degrees of freedom in the contact area, creating an ideal homogenously distributed bonding.

All the materials in the present study were considered homogenous, linearly elastic, and isotropic. The modulus of elasticity (E) and Poisson’s ratio (υ) were established based on the literature data listed in Table 1.

## 3. Results

The stress distributions over the dental components were computed according to the von Misses stress criterion. The von Misses stresses of the sound tooth model due to the protrusive movement and tearing function are presented in Figure 4. The effect of these forces at different angles and stress results in the buccal and palatal aspects are also shown. 

The stresses increased proportionally with the magnitude of the applied force concerning the protrusive movements and tearing function. The maximum von Misses stress values of the sound tooth model were found around 20 MPa and 59 MPa for loading with a 15 N 90° angle and 50 N 60° angle, respectively. 

The stress distributions in the PLV, resin cement, and an adhesive layer for both the butt joint and palatal chamfer preparations for different loading conditions are presented in Figure 5 and Figure 6. The effects of the elastic modulus of the cement on the stress distribution of restorations are also shown. In the FE model with the butt joint preparation, the maximum stress value of the restored tooth with a lower elastic modulus cement is 27 MPa, which is lower than the one with a high elastic modulus cement (33 MPa) for protrusive movement. While the maximum stress value of the restored tooth with the palatal chamfer preparation was found around 23 MPa for both low and high elastic modulus cement under the same loading condition. In the case of the tearing function, the maximum stress value of the restored tooth is found in the range of 69–72 MPa for lower and higher elastic modulus cement, respectively. For the palatal chamfer preparation at the same loading condition, maximum stresses are found lower, as 62 MPa independent of the elastic modulus. 

The maximum stress values induced in the PLV, resin cement (C), and adhesive layer (A) for the butt joint and palatal chamfer preparations are compared as a function of the cement elastic modulus in Figure 7. The maximum stress values of the PLV, resin cement, and adhesive layer were 16.8 MPa, 8.3 MPa, 8.3 MPa for a low elastic modulus, and 24.7 MPa, 17.3 MPa, 10.3 MPa for a high elastic modulus in the butt joint preparation under protrusive load, respectively. The maximum stress values of the PLV, resin cement, and adhesive layer were 11.4 MPa, 4 MPa, 6.7 MPa for low elastic modulus, and 9.8 MPa, 5 MPa, 6.6 MPa for high elastic modulus in the palatal chamfer preparation, respectively at the same loading condition. 

The maximum stress values of the PLV, resin cement, and adhesive layer were 69.9 MPa, 20 MPa, 19 MPa for low elastic modulus and 55.9 MPa, 33.7 MPa, 21.2 MPa for high elastic modulus in butt joint preparation under tearing function, respectively. The maximum stress values of PLV, resin cement, and adhesive layer were 48.6 MPa, 16.1 MPa, 19.1 MPa for low elastic modulus and 48.3 MPa, 18.9 MPa, 17.6 MPa for high elastic modulus in palatal chamfer preparation, respectively under same loading.

The average strain values obtained from 20 nodes in the cervical and buccal regions of each restoration type are shown in Figure 8. The strain results in the PLV, resin cement, adhesive layer, and prepared tooth (T) for various elasticity moduli of the cement are compared.

## 4. Discussion

Adhesive restorations dictate the use of materials that imitates the biomechanical behavior of the tooth structure in addition to providing aesthetics [26]. When performing laminate veneer restoration, the dentin may be exposed during tooth preparation due to the thin enamel, especially in the cervical area, even if the available tooth surface and guide-grooved burs with various diameters are taken into consideration [27,28]. This study considers the exposed tissues in the cervical area of the dentin, reflecting clinical conditions.

Many dental components, such as cortical bones [29], dentin [30,31], enamel [32], and periodontal ligaments, [33] show heterogeneous, anisotropic, and/or non-linear properties. However, measuring these properties in a clinical environment is not easy. Hence, these components were assumed to be linearly elastic, isotropic, and homogenous, and widely accepted mechanical properties in the literature were considered in the numerical models. It should be mentioned that despite the differences in local stress levels due to isotropic assumption, only minor consequences in the stress distribution take place in the qualitative analysis. It should also be noted that the linear elastic material assumption is appropriate for small deformations of brittle structures [16], and tetrahedral elements provide a good approximation modeling of complex and irregular tooth geometries [34]. To compute stresses in linear elastic materials hereby, the von Misses theory was used [35,36], which is a common failure criterion for restorative materials. FE results were verified with a convergence test by systematically comparing the stress distributions in critical areas with a reduced element size [34,37]. This increases the computational cost due to the increased number of elements and nodes but provides more reliable results.

Two types of loading conditions were considered to simulate the protrusive movements and tearing function. At this point, the maximum stresses of the sound and restored tooth were found to be 20 MPa for the applied 15 N force for protrusive movement and 59 MPa for the applied 50 N force for the tearing function. These values are in the same range as previous studies [15,38]. Accordingly, the main failure mechanisms of chipping, fracture, or debonding may occur at the incisal margin and cervical regions, which agrees with earlier studies determined that restoration failures occurred in the incisal and cervical regions (see Figure 4, Figure 5 and Figure 6) [30,39,40,41]. By examining Figure 5 and Figure 6, it is concluded that the stress values of PLV restorations with butt joint preparation were greater than those with palatal chamfer preparation independent of adhesive type. Hence, the sufficient tooth tissue supported in the model prepared with palatal chamfer may cause an increase in the bonding surface and support of the structure of the bonded porcelain at the incisal margin; thus, suggesting that the preparation method played a vital role in the stability of the restoration. Similarly, Jankar et al. [42] reported higher fracture resistance at porcelain laminate veneers prepared with palatal chamfer restorations compared with butt joint restorations. In addition, Li et al. [39] showed that lower stress distributions in the models were created with palatal chamfer preparation.

The stress distributions of the sound tooth model were similar to the stresses of the palatal chamfer restoration as shown in Figure 4, Figure 5 and Figure 6. This is mainly due to the increased volume of the PLV prepared in the palatal edge and the adhesive surface area expansion, which increases restoration resistance to shear stresses. In clinical use, porcelain laminate veneers can break due to the magnitude of bite loads. However, cracking in the cement and porcelain layer and chipping are more likely to occur because of low and continuous bite loads [43]. It was reported in earlier studies that the porcelain material showed a higher strength to compressive forces but could not withstand tensile forces without fracture. Tensile forces were concentrated in the palatal concavity of the upper central teeth. While restoring these teeth with a PLV by creating a chamfer preparation, which included the incisal margin, without extending to the palatal concavity avoids the tensile forces during mastication at the palatal end of the porcelain [38]. Sorrentino et al. [40] reported that the PLV restorations showed greater stress values and lower deformations than those in the sound tooth model. This behavior was mainly due to the higher elasticity modulus of the porcelain material in comparison to the enamel at the same location in the sound tooth. 

As can be seen in Figure 5 and Figure 6, when forces were applied at angles of 90° and 60°, stress values in the buccal incisal margin in both the butt joint and palatal chamfer design preparations were higher than those in the same region of the sound tooth. According to these findings, one could indicate that the area with the highest failure risk in the PLV restoration was the buccal incisal margin. This mechanism could also be the reason for cohesive fractures in the incisal margins of the PLV restorations observed in clinical practice. Moreover, relatively higher stress concentrations that occurred in the incisal margins of PLV were mentioned in the previous studies [30,44,45]. Compared with the stress concentration in the buccal and cervical areas of the porcelain layer, the stresses in the adhesive cement layer were relatively low.

Stresses in the resin cement of the PLV restoration with the palatal chamfer design were lower than that of the butt joint preparation design. This observation gave rise to the idea that the increased volume of adhesive cement used in PLV restoration and the associated increase in the porcelain-cement ratio was significant, which was consistent with previously reported results [46].

The mechanical properties of restorative materials affect the clinical performance of restoration regarding strength and longevity. Under functional loads, stresses are inclined to be concentrated at interfaces between the structures with different mechanical behavior [47]. The maximum stress distribution was concentrated on the porcelain layer and decreased toward the cement layer up to the dental tooth structure. The stress distributions in both restoration designs increased in parallel to the elasticity modulus of the resin cement (e.g., see Figure 5, Figure 6 and Figure 7). Higher stiffness in the adhesive layer increased the rigidity, inducing greater stress distributions on tooth structures. These findings were also similar to the assessment reported by Spazzin et al. [48], which showed an increased risk of debonding caused by higher stress values due to the use of adhesive cement with high elasticity modulus. Furthermore, as can be seen in Figure 5, Figure 6 and Figure 7, the change in stress distributions in the butt joint preparation due to different elasticity moduli was found to be greater than that of the palatal chamfer preparation. Our results also showed that the stress values in the PLV, resin cement, and adhesive layers slightly increased proportionally with the increase in the elasticity modulus of the resin cement. As the elastic modulus of the resin cement increased, the stress values on the PLV decreased in all PLV-restored models except the butt joint preparation with the 90-degree mechanical loads. It can be shown that the loading state, preparation of geometric configuration, and material properties are effective on the stress change. The increase in the elastic modulus of the cement caused less stress accumulation in the porcelain layer. This finding is accompanied by similar unit displacements of the porcelain with the high elastic modulus luting cement at strain values compared to the low elastic modulus cement (Figure 8).

While stress distribution occurred over rigid structures, withstanding forces and deformations occurred in various directions. The greater elastic modulus shows a higher resistance to deformation, resulting in lower strain values [12,13,14,30,40]. This is in line with our results. As shown in Figure 8, the highest strain values were found in the resin cement and adhesive layers, while the lowest strain values were found in the PLV in both loading conditions. The strain values in the cervical and buccal regions were qualitatively similar for different loading conditions (Figure 8). The overall strain in each component was proportionally increased with the increased magnitude of the applied force. Independent of the loading conditions, the strain values in both the buccal and cervical regions of the porcelain and resin cement were similar for the two different elastic moduli of resin cement. For both the PLV restorations with low elastic modulus, the cement layer exhibits similar strain values as the adhesive and dental tissues relative to the high elastic modulus cement in the cervical region. Failure or debonding may occur in the restored tooth because of the dissimilar displacement behavior of the materials under mechanical loading. Furthermore, the increase in the elastic modulus may reveal different deformation characteristics, potentially producing problems with bonding between the adhesive layer and cement. Therefore, this directly affects the interfacial combination, where problems regarding bonding may occur in the restoration. It also dictated whether the bonding problems in the interface arose between the porcelain and resin cement or between the adhesive cement and tooth. Due to insignificant differences in the strain values of the adhesive layer in models prepared with a palatal chamfer, differences in strain values between the resin cement and the adhesive layer were also found to be relatively low. Therefore, fewer strength problems are anticipated. 

From the clinical point of view, the higher elastic modulus resin cement showed similar biomechanical behavior with the porcelain material under the actual loading forces. In comparison, the lower elastic modulus resin cement showed similar biomechanical behavior with the dentin tissue. The difficulties of adhesive bonding to dentin are reported in the literature. Due to the diversity of the characteristics of dentine and enamel, the bonding problems differ. The dentine substrate varies from the enamel in the tubular structure, containing lower inorganic content and having outward intratubular fluid movement [49]. Restorations are expected to behave mechanically similarly to dental structures [50]. Due to dental structure loss, choosing a restorative material, such as enamel with a high elastic modulus or dentin with a low elastic modulus, is essential in terms of restoration longevity since they show similar behavior with the dental structure under stress. Since resin cement with a low elastic modulus shows similar stress distribution and strain (displacement) values as dentin, this should be considered for the continuity of the adhesive bond in preparations where dentin is exposed. Thus, these findings give the idea that if the clinician has doubts about adhesion quality, especially in the clinical cases where dentin-exposed tooth preparations are performed, a lower elastic modulus resin cement might be eligible due to the biomechanical behavioral similarity with the dentin tissue. 

Porcelain and resin cement showed strain values approaching each other with the increasing elasticity modulus of the adhesive cement used in the cervical and buccal areas of both models. When cement with a high elasticity modulus was used, the deformation differences between the porcelain and cement layers were low (Figure 8). This may prevent against inconvenient fractures by causing less stress on the porcelain layer on the incisal edges with high-stress areas. Discrepancy occurrence was observed in terms of strain values mainly between the adhesive layer and prepared tooth tissue with the increasing elasticity modulus of adhesive cement in the cervical area of models equipped with a butt joint and palatal chamfer under both loading conditions. When the cervical margin was examined in terms of debonding, this study showed that adhesive cements with a low elasticity modulus should be used due to the little difference in strain values between the adhesive layer and the prepared tooth. The risk of clinical failure in different regions associated with variations in elastic modulus should be taken into consideration.

## 5. Conclusions

Within the limitations of this study, the following conclusions were drawn:PLV restorations with the palatal chamfer involving the incisal margin design results in biomechanical responses closer to that of a sound tooth compared to restorations using butt joints.Regardless of the magnitude and angle of the load and the elasticity modulus of the resin cement, the PLV restorations with butt joint preparation showed higher stress distributions compared to those with palatal chamfer preparation.An increase in the elasticity modulus of the resin cement induced slightly larger stresses on the adhesive layer, tooth tissues, and restorative materials.

## Figures and Tables

**Figure 1 materials-16-02378-f001:**
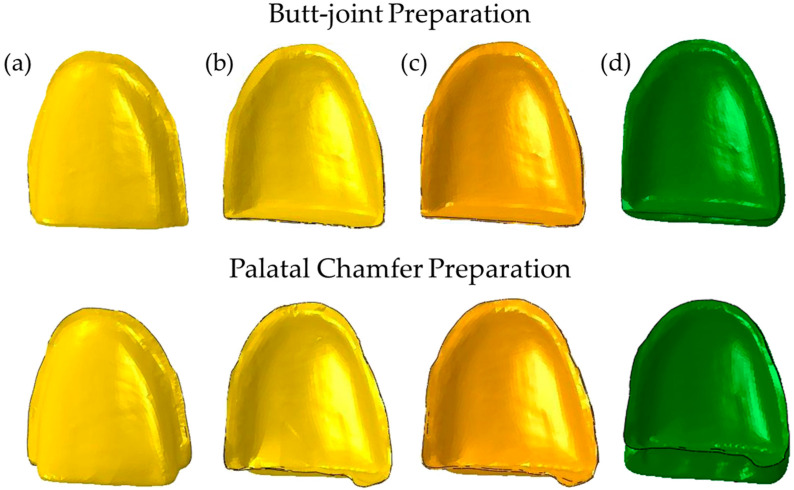
Butt joint and palatal chafer preparations: (**a**) adhesive layer in buccal aspect, (**b**) adhesive layer in palatal aspects, (**c**) resin cement in palatal aspects, and (**d**) laminate veneer in palatal aspects.

**Figure 2 materials-16-02378-f002:**
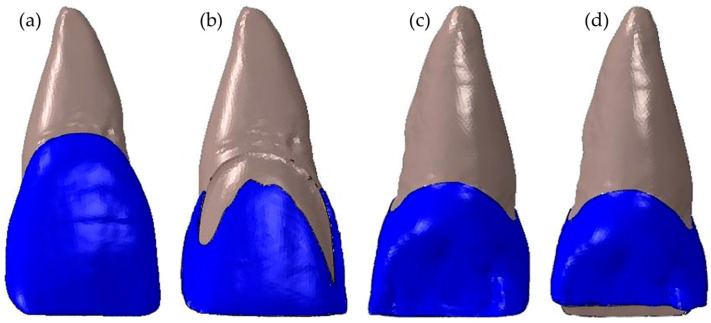
Solid models of (**a**) the sound tooth, (**b**) the butt joint prepared tooth in buccal aspect, (**c**) the butt joint prepared tooth in palatal aspect, and (**d**) the palatal chamfer prepared tooth in palatal aspect.

**Figure 3 materials-16-02378-f003:**
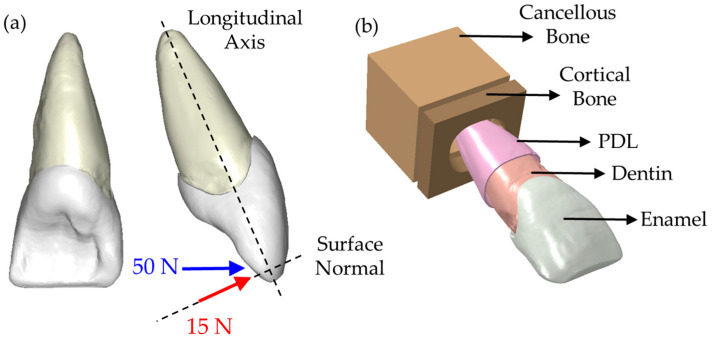
(**a**) Applied forces and (**b**) CAD model of the sound tooth.

**Figure 4 materials-16-02378-f004:**
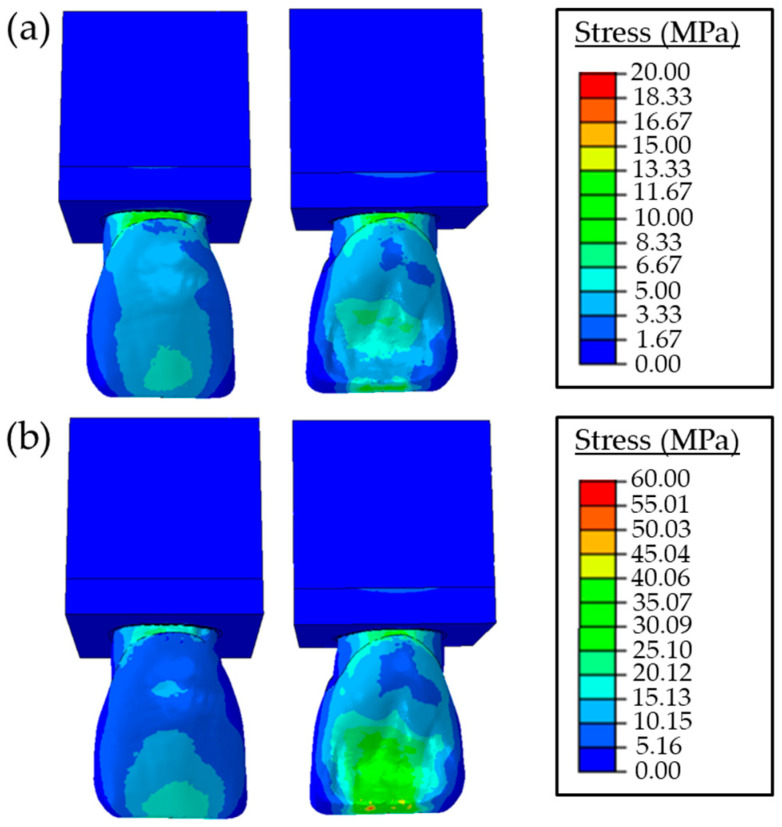
Stress values of the sound tooth with (**a**) the protrusive movement due to a horizontal force of 15 N with 90° angle and (**b**) the tearing function due to an oblique force of 50 N with 60° angle.

**Figure 5 materials-16-02378-f005:**
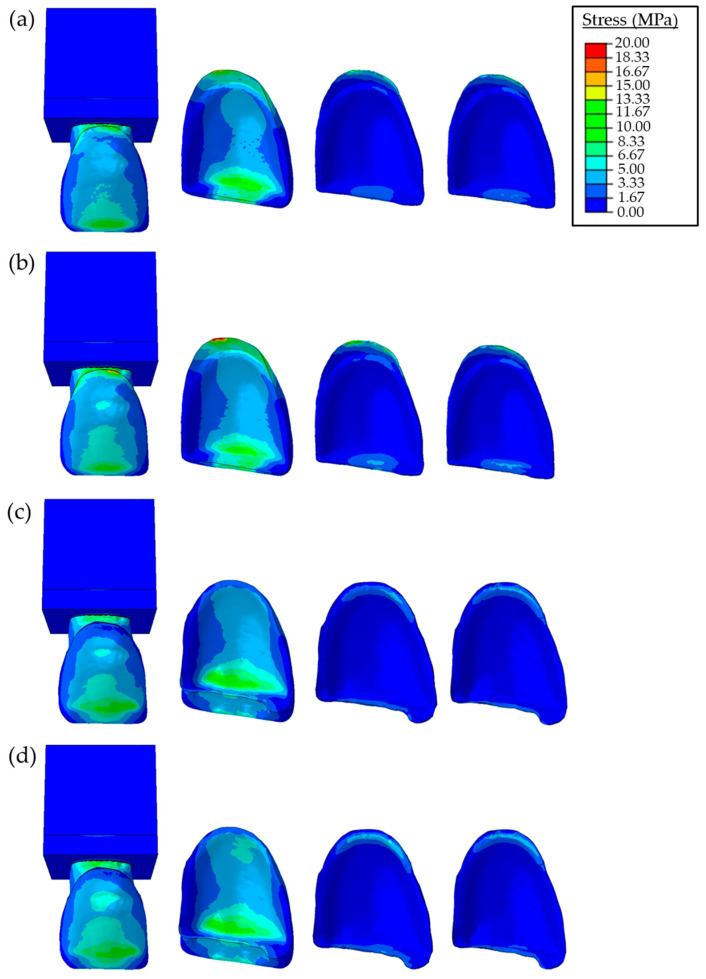
Stress distributions due to a horizontal force of 15 N with 90° angle in the restored teeth, porcelain laminate veneer, adhesive cement, and adhesive layer, respectively. Butt joint preparation using (**a**) low modulus, (**b**) high modulus resin cement, and palatal chamfer restoration using (**c**) low modulus and (**d**) high modulus resin cement.

**Figure 6 materials-16-02378-f006:**
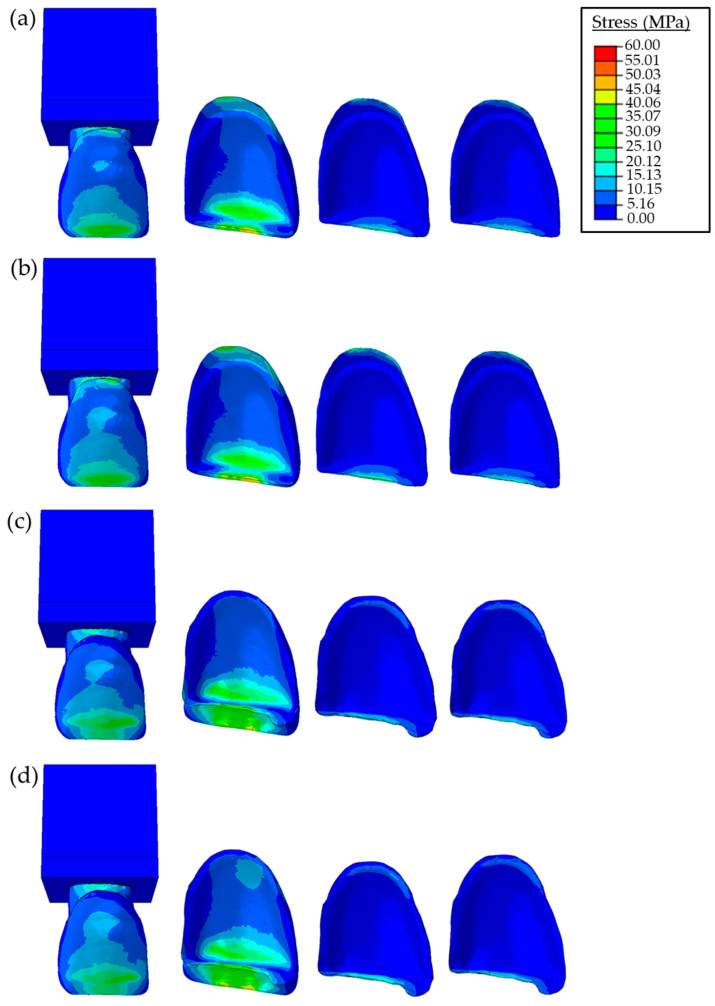
Stress distributions due to an oblique force of 50 N with 60° angle in the restored teeth, porcelain laminate veneer, resin cement, and adhesive layer, respectively. Butt joint preparation using (**a**) low modulus and (**b**) high modulus resin cement, and palatal chamfer restoration using (**c**) low modulus and (**d**) high modulus resin cement.

**Figure 7 materials-16-02378-f007:**
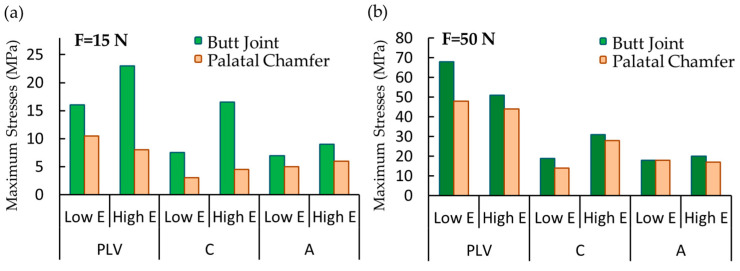
The maximum stress values in the porcelain laminate veneer (PLV), resin cement (C), and adhesive layer (A) for butt joint and palatal chamfer preparations in response to (**a**) a 15 N force for protrusive movement and (**b**) a 50 N for the tearing function.

**Figure 8 materials-16-02378-f008:**
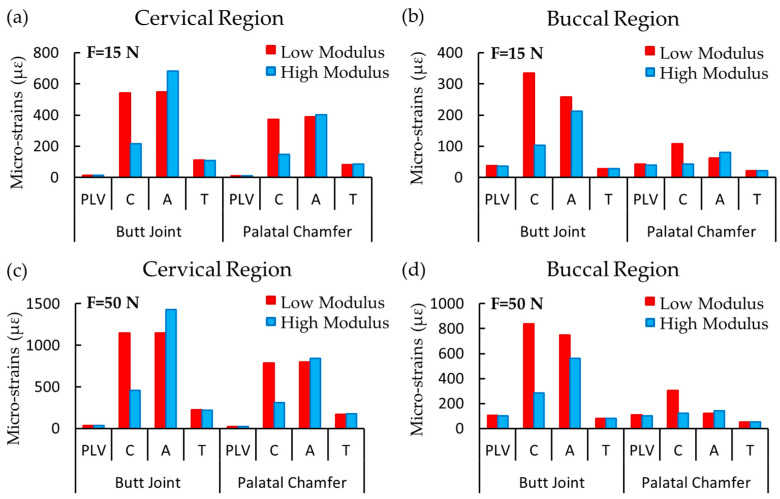
The average strain values in the porcelain laminate veneer (PLV), resin cement (C), adhesive layer (A), and prepared tooth (T) (**a**) in cervical region, (**b**) in buccal region due to the 15 N force (F) of protrusive movement, (**c**) in cervical region and (**d**) in buccal region due to 50 N force of tearing function.

**Table 1 materials-16-02378-t001:** Material properties of the tooth layers and restorative materials.

Structures and Materials	Elasticity Modulus (MPa)	Poisson’s Ratio
Enamel [5]	84,100	0.33
Dentin [5]	18,600	0.32
Periodontal ligament [24]	68.9	0.45
Spongious bone [24]	1370	0.30
Cortical bone [24]	13,700	0.30
Adhesive [25]	5000	0.29
Cement with low modulus Variolink Veneer, Ivoclar Vivadent AG, Schaan, Liechtenstein [*]	4500	0.24
Cement with high modulus Ena Cem HF, Micerium S.p.A., Avegno (GE), Italy [*]	12,850	0.24
Porcelain laminate veneer IPS e.max Press, Ivoclar Vivadent AG, Schaan, Liechtenstein [*]	95,000	0.23

* Values provided by manufacturers.

## Data Availability

Not applicable.

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
