# Peer review of "Biomechanical Behavior Evaluation of Resin Cement with Different Elastic Modulus on Porcelain Laminate Veneer Restorations Using Micro-CT-Based Finite Element Analysis"

_materials, 2023, doi:10.3390/ma16062378_

Round 1

Reviewer 1 Report

The study presented for the review describes the porcelain laminate veneer. The introduction clearly presents the authors’ motivation. The “preparation” section is written cohesively. I would suggest evaluating the contact angle testing which gives more information about surface energy, and adhesion strength. The particular comments on each of the parts are listed below:

1. The value of tested samples for the stress analyses is missed.

2. The error bars are missing.

3. It may be interesting to show the statistic p-value.

4. The contact angle analysis and the surface free energy are two important factors if the study investigated the interaction between two materials. I highly recommend testing it.

5. Line 246: Please specify, which features or dental compounds are described as heterogeneous, anisotropic, and non-linear.

6. It does not discuss how the contact area size between the teeth and the veneer influences the adhesion strength.

7. The article's title says about micro-CT imaging. The comparison with the other techniques is not discussed.

 8. Table 1 shows the data from 2005 and 2006. Are they the current ones? 

Author Response

We appreciate the Editor and Reviewers’ detailed comments and constructive suggestions for the manuscript. Response letter is attached at the end of the manuscript.

Reviewer 2 Report

Manuscript ID: materials-2218026

Title: Biomechanical behavior evaluation of resin cement with different elastic modulus on porcelain laminate veneer restorations using micro-CT-based FEM

Reviewer Comments: Authors must respond to these remarks in order to improve their paper. The article can be reconsidered for publication after a major revision only.

1.       The author must include a statement about the research's novelty.

2.       How does the elasticity of resin cement affect how porcelain laminate veneer restorations work?

3.       What is the impact of the resin-cement elastic modulus on the stress distribution in the restoration?

4.       How does the modulus of the resin cement affect how the restoration fits at the edges?

5.       What is the effect of the elastic modulus on the microleakage and bond strength of resin cement to the tooth structure and porcelain veneer?

6.       How does the elastic modulus of the resin cement affect the longevity of the restoration in different clinical scenarios?

7.       What is the effect of the resin cement elastic modulus on the fracture toughness of the restoration?

8.       How can micro-CT-based FEM be used to accurately measure how different elastic moduli of resin cements affect porcelain laminate veneer restorations?

Author Response

(The authors gave the same response as above.)

Reviewer 3 Report

This is a well-written manuscript.

Stress at the laminated interface, i.e. bonding interface, should be evaluated and discussed. Discussion on the bonding strength of the veneer and the strength of the cement should be included. Most possible failure locations in the structure can be pointed based on the material or interface strengths.

 Although this study is not aiming to simulate the failure event, in table 1, the tensile/compressive strength of each material can be included to envision the possible failure location.

In figure 8, using a milli-strain or microstrain would be better in the plot to avoid many zeroes.

The resin cement thickness is not uniform in real life and it is normally much thinner in the margin region than in the body region. It can be less than 15 micrometers at the margin. With a high-resolution optical scanner or micro CT, the such deviation can be easily measured.

Author Response

(The authors gave the same response as above.)

Round 2

Reviewer 2 Report

the revised manuscript can be accepted.